# Translation and cross-cultural adaptation of the Persian version of inflammatory bowel disease-fatigue (IBD-F) self-assessment questionnaire

**Mohammad Reza Hatamnejad[1], Marzieh Shirvani[2], Mohamad Amin Pourhoseingholi[3], Hedieh Balaii[1], Shabnam Shahrokh[1]\*, Hamid Asadzadeh Aghdaei[1], Amirhosein Koolaeian[1], Makan Cheraghpour[1]\***

1 Research Institute for Gastroenterology and Liver Diseases, Basic and Molecular Epidemiology of Gastrointestinal Disorders Research Center, Shahid Beheshti University of Medical Sciences, Tehran, Iran, 2 Student of Research Committee, Shiraz University of Medical Sciences, Shiraz, Iran, 3 Gastroenterology and Liver Diseases Research Center, Research Institute for Gastroenterology and Liver Diseases, Shahid Beheshti University of Medical Sciences, Tehran, Iran

\* shabnamshahrokh@gmail.com (SS); bio_makan@yahoo.com (MC)

**Data Availability Statement:** The datasets used and/or analyzed during the current study are

## Abstract

### Background

This study appraises the psychometrics properties of the Inflammatory bowel disease-fatigue (IBD-F) Persian version questionnaire.

### Methods

The original IBD-F questionnaire was translated into the Persian version in a standard forward-back manner. The validation was performed through the face, content, and construct validity. Fifteen experts scored each item's necessity on an ordinal Likert scale of three; then, the content validity ratio was calculated using the Lawshe formula. Eight judges from pre-defined panel rated each item on an ordinal 4-point Likert scale concerning its relevancy, clarity, and simplicity for individual-CVI calculation. The mean individual-CVI was considered as the Scale-CVI for each domain. Twenty lay experts (selected from the target population) were asked to express their opinion on each item's importance by scoring on a 5-point Likert scale; subsequently, face validity was determined by the impact score formula. The questions that had minimum values of CVR, CVI, and impact score were retained in the final version of the questionnaire for reliability evaluation. Construct validity was evaluated via Confirmatory Factor Analysis. Internal consistency and test-retest reliability were checked using Cronbach's α and intraclass correlation coefficients (ICC). Fifty-four patients recruited based on inclusion and exclusion criteria to participate in the reliability analysis.

### Results

All the questions received the qualified values of CVR (exceeding 0.49 points) and impact score (more than 1.5 points) and were retained in the questionnaire; however, revisions

available from Figshare repository, via DOI: 10. 6084/m9.figshare.23292446.

**Funding:** The author(s) received no specific funding for this work.

**Competing interests:** The authors have declared that no competing interests exist.

**Abbreviations:** IBD, Inflammatory bowel disease; CD, Crohn's disease; UC, Ulcerative colitis; QoL, Quality of life; BFI, Brief Fatigue Inventory; MFI, Multifactorial Fatigue Inventory; FACIT-F, Functional assessment of chronic illness therapy-fatigue; FQ, Fatigue questionnaire; IBD-F, Inflammatory Bowel Disease-Fatigue; CVR, Content validity ratio; CVI, Content validity index; I-CVI, Individual-CVI; S-CVI, scale-CVI; CFI, Comparative fit index; RMSEA, Root mean-square error of approximation; CFA, Confirmatory Factor Analysis; ICC, Intraclass correlation coefficients.

were made for questions with a CVI 0.7–0.9 for clarity and simplicity. The result demonstrated relative goodness CFA and proper internal consistency, as Cronbach's α coefficient was 0.964 for the test (0.845 and 0.963 for the first and second part of the questionnaire, respectively (and 0.888 for the re-test (0.793 and 0.876 for the first and second section of the questionnaire, respectively). The ICC values between test and re-test for the first and second part and the whole questionnaire were obtained as 0.904, 0.922, and 0.921, respectively.

## Conclusion

The Persian version of the IBD-F questionnaire was valid and reliable; thus, an appropriate scale was deemed to measure fatigue (severity, frequency, and impact on daily activities).

## Background

Inflammatory bowel disease (IBD), consisting of Ulcerative colitis (UC) and Crohn's disease (CD), is appointed as an idiopathic inflammatory disorder consequence of host-microbial interaction [1]. Despite intestinal symptoms, systemic manifestations are common concerns among patients with IBD. Fatigue, as one of the most prevalent presentations, rigorously impresses the quality of life (QoL) and exists in 41% of cases [2]. This multifactorial symptom can be defined in various dimensions like general weakness, inability to initiate movement, difficulty keeping on with an activity, emotional instability, and memory lapse [3]. Numerous factors contribute to fatigue in IBD patients, such as disease activity index [4–6], psychosomatic stresses [6] and mental disorders like anxiety and depression [2,4], IBD-related medications such as Azathioprine, Anti-TNF-α [7], Corticosteroids, and Methotrexate [8,9], and alterations in the gut-brain axis [10].

Fatigue is determined as the level of affairs individuals can carry out; thus, measuring fatigue provides an opportunity to know how effective the patient is and whether necessary intervention is needed. Several scales, settled to measure fatigue in chronic diseases, are accessible for patients with IBD. The most common scales that represent different approaches for measuring fatigue are the Brief Fatigue Inventory (BFI) which compromises a rapid fatigue evaluation [11], Multifactorial Fatigue Inventory (MFI) which evaluates fatigue in five scopes (namely general, physical, motivational, and mental) [12], Functional assessment of chronic illness therapy-fatigue (FACIT-F) [13], and Fatigue questionnaire (FQ) [14]. Some of them have been validated in the IBD population [13,14]; however, there was a need for a more comprehensive questionnaire that more accurately examines the impact of fatigue on the daily activities of IBD patients.

In 2014, Czuber-Dochan. et al. proposed a self-assessing questionnaire, Inflammatory Bowel Disease-Fatigue (IBD-F), to assess fatigue in patients with IBD, including mental and physical items [15]. The original version of the IBD-F questionnaire comprises three sections. The first part, including five questions, generally assesses the fatigue within the previous two weeks. The subsequent thirty questions construct the second part and appraise the effect of fatigue on daily affairs in the last two weeks. The third part, comprising the five non-scoring open-ended items, is designed to consult the health professions about other possible sources and properties of fatigue.

There are considerable advantages for IBD-F questionnaire that persuade researchers and physicians to select this questionnaire between fatigue measuring scales. For instance, the

FACIT questionnaire is designed for patients with chronic illness and has been used in several studies for IBD patients [16,17]; it is a 13-item patient-reported fatigue measuring with a 7-day recall period. Meanwhile, the IBD-F questionnaire has specifically been schemed for IBD patients since it contains 35 activity-related items. Furthermore, the other benefit of the IBD-F questionnaire is extended coverage of recall period (measuring the fatigue within the past two weeks) and five final open-ended questions for counseling purposes.

A significant advantage of IBD-F is that it is purposed to concomitantly appraise the severity and the effect of fatigue on patient well-being [18]. Transcultural adaptation of the IBD-F questionnaire in Iranian society has not been specified; hence, this study aimed to appraise the psychometrics properties of the Persian version of the IBD-F questionnaire.

## Methods

All patients gave written informed consent before the study. The Shahid Beheshti University Medical Sciences ethics committee approved the study protocol (ethics code: IR.SBMU. RIGLD.REC.1401.005).

### 1. Translation

Before the validation process, the original IBD-F questionnaire translated in a standard forward-back manner [19]. The authors of the original version of the questionnaire were contacted and their consent obtained. Two proficient translators, who spoke Persian natively and were fluent in English, with prior experience in dealing with IBD patients, independently translated the original questionnaire into the two Persian versions (*forward translation*). Then, an expert panel (a gastroenterologist, a dietitian, a medical doctor, and a nurse) compared these versions and rectified their discrepancies; the final version approved. In the next step, two independent translators, native English speakers and highly dominant in the Persian language, accomplished *back-translation* (from Persian to English). The panel members compared the original issue with the accepted final English version [15]; all the necessary changes were applied, and a definite questionnaire prepared for measuring the validity and reliability [19].

### 2. Validation

Validity should be assessed to define how well the IBD-fatigue questionnaire is translated and transfer the concept. For this purpose, face, content, and construct validity were selected to validate the questionnaire [20].

**2.1 Content validity.** The expert panels' viewpoints determine the research instrument's content validity. Experts were qualified based on previous research experience or clinical expertise in the field of IBD. During the personal visit, a trained interviewer explained the purpose of the investigation and how experts should score the items (Table 1). In the first step, the content validity ratio (CVR) was computed to specify the necessity of each question; in other words, it distinguished the retention or exclusion of each questionnaire item. Fifteen judges scored each item from 1 to 3 with a three-degree range of "not necessary, useful but not essential, essential" Table 1. Then, CVR was calculated from the Lawshe formula [21] as $(Ne-N/2)/(N/2)$, in which (N) is the total number of judges and (Ne) indicates the number of them who selected number three (assumed item as essential). Based on the Lawshe table [21], a minimum CVR value of 0.49 (when there are fifteen panelists) is required to maintain the item in the questionnaire. We omitted items, which did not meet the minimum CVR value, and calculated the content validity index (CVI) for the final questionnaire.

**Table 1. Panelists and lay experts scoring system for calculating the content validity for each item.**

| Scores / Characteristics | 1 | 2 | 3 | 4 | 5 |
|---|---|---|---|---|---|
| Necessity | Not necessary | Useful but not essential | Essential | - | - |
| Relevancy | Not relevant | Item needs some revision | Relevant but needs minor revision | Very relevant | - |
| Clarity | Not clear | Item needs some revision | Clear but needs minor revision | Very clear | - |
| Simplicity | Not simple | Item needs some revision | Simple but needs minor revision | Very simple | - |
| Importance | Unimportant | Slightly important | Relatively important | Important | Very important |

Lynn [22] recommended a range of 3(minimum) to 10(maximum) panelists for quantifying CVI; thus, eight of them were asked to score on an ordinal Likert scale of four to each item concerning relevancy [22], clarity, and simplicity [23]. The validity scoring system, which is provided in Table 1, was as follows:

1. **For relevancy:** 1 = not relevant, 2 = item needs some revision, 3 = relevant but needs minor revision, 4 = very relevant.

2. **For clarity:** 1 = not clear, 2 = item needs some revision, 3 = clear but needs minor revision, 4 = very clear.

3. **For simplicity:** 1 = not simple, 2 = item needs some revision, 3 = simple but needs minor revision, 4 = very simple.

CVI was measured for each item (individual-CVI) and the whole instrument (scale-CVI). The number of experts giving a three or four score to each item, divided by the total number of the experts, constitutes the CVI-I; in addition according to the average of I-CVIs, S-CVI was calculated [24]. I-CVI values over 0.79 were desirable, while values between 0.7 and 0.79 displayed that item needs revision. Items with values lower than 0.7 should be eliminated from the study [25].

**2.2 Face validity.** A face validity test supports content validity, which means it examines whether an instrument appears valid for subjects or participants. It is mainly related to the appearance and measure of participants' agreement with items and the wording of the instrument to achieve the research goal [20]. Thus, 20 lay experts (chosen from the target population) read the questionnaire items carefully online and declare their opinion about each item's importance by donating a 5-point Likert scale. The lay experts' scoring system was as follows: 5 = very important, 4 = important, 3 = relatively important, 2 = slightly important, and 1 = unimportant Table 1. Face validity was evaluated through the item impact score formula (*frequency* * *importance*), in which "*frequency*" is attributed to the percent of patients who consider the question important or very important, and "*importance*" is the mean importance score of the item. Questions with an impact score equal to or more than 1.5 were retained, and rest were removed.

**2.3. Construct validity.** In order to evaluate the construct validity of IBD-F questionnaire, Confirmatory Factor Analysis (CFA) was performed using LISREL software (version 8); Chi-Square, Comparative fit index (CFI), and Root mean-square error of approximation (RMSEA) were reported to evaluate the fitness of observed data to the model fitness [26,27].

## 3. Reliability

**3.1 Sample size calculation and patient selection.** The minimum required sample size, based on alpha-error: 0.05, beta-error: 0.05, power: 0.95, and effect size: 0.83 [15] was calculated by G* Power software [28] as 12 patients; Subsequently, 54 patients participated in

reliability analyses. Moreover, in the first original IBD-F version, a test-retest analysis was conducted on 36 candidates [15].

Based on providing the consent form and owning the following criteria, patients were selected and included: at least 18 years, verified diagnosis of IBD, and ability to read and realize the Persian language. During the outpatient visit to the gastroenterology clinic of Taleghani hospital (a university-affiliated tertiary medical center), the patients filled out the IBD-F questionnaire and an inquiry related to their socio-demographic and medical data. Disease activity was measured via the Harvey–Bradshaw and Simple Clinical Colitis Activity Indexes in CD and UC patients [29,30]. Four scores or below were considered the remission stage; in opposition, higher values indicated the active phase of the disease.

**3.2 Internal consistency and reproducibility.** As the last five items are open-ended descriptive questions and not involved in fatigue score calculation, they were not considered for reliability checking. However, their retention in the final questionnaire version, for consulting purposes, was considered in the validity evaluation by the panelists and lay experts.

Using Cronbach's α coefficient, internal consistency was tested for parts I, II, and the whole instrument. After 14 days, all patients were recalled to attend the outpatient clinic for a second assessment. They were excluded if any change associated with disease activity was observed within the interval between the test and re-test. Replies to the items were compared via test-retest reliability using the intraclass correlation coefficients (ICC). However, before all analyses, fatigue scores should be adjusted; since instrument section two, contains the questions with a "non-relevant" option response. Adjusted fatigue score was calculated via the following formula: maximum possible score * [total score / (max possible score)—(number of questions with non-relevant answer*4)].

## 4. Statistical analysis

Sample size calculation was done via G* Power version 3.1.9. CFA was performed using LISREL (version 8); Other statistical analyses were accomplished via the R statistical package, version 3.3 (R Development Core Team 2017). A significance level of 0.05 was considered for all tests.

## Results

### Validation

**1 Content validity.** In the first step, fifteen judges scored each item to calculate CVR by the Lawshe formula, which resulted in specifying the necessity of each question. A minimum CVR value of 0.49 (with fifteen panelists) was required to keep the item; finally, all questions got the eligible point and were retained in the questionnaire Table 2.

Eight panelists scored based on an ordinal Likert scale of four to each item concerning its relevancy, clarity, and simplicity. In the clarity index, two questions (Q4 and Q23) received 0.75 points; the rest got higher scores than 0.79. Five questions (Q2, Q3, Q4, Q29, and Q38) in the simplicity domain got 0.75 points, and others achieved higher than 0.79. Concerning relevancy, all questions got the required score except for one item (Q11), which gained a 0.75 mark. S-CVI (Average of I-CVIs) for clarity, simplicity, and relevance were achieved at 0.878, 0.903, and 0.896, respectively. Questions with 0.7–0.79 score, were appropriately modified to achieve the needed score (I-CVI for each item) in re-examining and remained in the questionnaire Table 2.

**2 Face and construct validity.** Twenty lay experts read the questionnaire and donated an importance score to each item. Face validity was evaluated through the item impact score formula. The least and highest achieved impact score were 1.70 (for Q24) and 3.50 (for Q20),

**Table 2. Validity of the Persian version of IBD-F questionnaire.**

| Validation Scales / Items | Face Validity | Content Validity | | |
|---|---|---|---|---|
| | Impact score | CVR | Individual-CVI | |
| | | (Necessity) | Clarity | Simplicity | Relevancy |
| Question 1 | 2.05 | 0.733 | 0.875 | 1 | 1 |
| Question 2 | 2.95 | 0.6 | 0.875 | **0.750** | 1 |
| Question 3 | 1.85 | 0.6 | 0.875 | **0.750** | 1 |
| Question 4 | 2.10 | 0.733 | **0.750** | **0.750** | 1 |
| Question 5 | 2.45 | 0.6 | 0.875 | 0.875 | 1 |
| Question 6 | 2.40 | 0.733 | 0.875 | 0.875 | 0.875 |
| Question 7 | 1.80 | 0.733 | 1 | 0.875 | 0.875 |
| Question 8 | 2.05 | 1 | 1 | 1 | 1 |
| Question 9 | 3.10 | 0.866 | 0.875 | 0.875 | 1 |
| Question 10 | 3.40 | 0.6 | 0.875 | 1 | 0.875 |
| Question 11 | 2.85 | 0.6 | 0.875 | 0.875 | **0.750** |
| Question 12 | 2.00 | 0.733 | 0.875 | 1 | 0.875 |
| Question 13 | 1.85 | 0.733 | 0.875 | 1 | 0.875 |
| Question 14 | 2.25 | 0.733 | 0.875 | 1 | 0.875 |
| Question 15 | 2.25 | 0.866 | 0.875 | 1 | 0.875 |
| Question 16 | 1.75 | 0.6 | 0.875 | 0.875 | 0.875 |
| Question 17 | 2.60 | 0.866 | 0.875 | 1 | 0.875 |
| Question 18 | 3.20 | 0.866 | 0.875 | 1 | 0.875 |
| Question 19 | 3.00 | 0.866 | 0.875 | 1 | 0.875 |
| Question 20 | 3.50 | 0.866 | 0.875 | 0.875 | 0.875 |
| Question 21 | 2.10 | 0.733 | 0.875 | 0.875 | 0.875 |
| Question 22 | 3.35 | 0.6 | 0.875 | 0.875 | 0.875 |
| Question 23 | 2.00 | 0.6 | **0.750** | 0.875 | 0.875 |
| Question 24 | 1.70 | 0.733 | 0.875 | 0.875 | 0.875 |
| Question 25 | 2.25 | 0.6 | 0.875 | 0.875 | 0.875 |
| Question 26 | 2.45 | 0.6 | 0.875 | 0.875 | 0.875 |
| Question 27 | 2.70 | 0.6 | 0.875 | 0.875 | 0.875 |
| Question 28 | 1.75 | 0.733 | 0.875 | 0.875 | 0.875 |
| Question 29 | 1.90 | 0.6 | 0.875 | **0.750** | 0.875 |
| Question 30 | 1.80 | 0.6 | 0.875 | 0.875 | 0.875 |
| Question 31 | 2.10 | 0.733 | 0.875 | 1 | 0.875 |
| Question 32 | 3.15 | 0.733 | 0.875 | 1 | 0.875 |
| Question 33 | 2.75 | 0.866 | 0.875 | 1 | 0.875 |
| Question 34 | 2.45 | 0.733 | 1 | 1 | 0.875 |
| Question 35 | 2.50 | 0.866 | 0.875 | 0.875 | 1 |
| Question 36 | 3.05 | 0.733 | 0.875 | 0.875 | 0.875 |
| Question 37 | 2.45 | 0.6 | 0.875 | 0.875 | 0.875 |
| Question 38 | 3.70 | 0.6 | 0.875 | **0.750** | 0.875 |
| Question 39 | 2.90 | 0.733 | 0.875 | 0.875 | 0.875 |
| Question 40 | 3.10 | 0.733 | 0.875 | 0.875 | 0.875 |
| | | Scale-CVI/Ave | 0.878 | 0.903 | 0.896 |

CVR, Content validity ratio; CVI, Content validity index.

Values lower than the accepted cutoff are illustrated in bold.

respectively; thus, all the questions got the eligible impact score (more than 1.5) and were retained in the final version Table 2. The final Persian version of the IBD-F scale is provided as the S1 File. In CFA analysis, Chi-Square, CFI, and RMSEA were reported to be 853.362, 0.8, and 0.1, respectively. These indexes show the relative property (borderline results) of observed data to the model fitness.

## Reliability

Fifty-four patients were consecutively recruited based on providing the consent form and owning determined criteria to participate in the test stage of reliability analysis. Data of these participants are illustrated in Tables 3 and S1.

The population comprised 52% UC patients against 48% CD patients. They were mainly women (57%), young (mean age of 36.5 years), and possessed the active form of the disease (65%), and with a normal range of body mass index (mean 23.6 Kg/m2). Patients had developed IBD for an average of 5 years; moreover, they were sparingly consuming alcohol (13%) and smoking (9%). Beyond half of them (67%) didn't report comorbidities; however, other gastrointestinal disorders (except IBD) were confirmed in 11%. TNF inhibitors, followed by

**Table 3. Demographic and clinical characteristics of the participants.**

| Variables (number (%) or mean ± SD) | Study Groups | | |
|---|---|---|---|
| | All Patients (N = 54) | CD patients Group (N = 26, 48%) | UC Patients Group (N = 28, 52%) |
| Age (year) | 36.5 ± 12.3 | 37.8 ± 14 | 35.2 ± 10.6 |
| Gender | | | |
| Women | 31 (57%) | 13 (50%) | 18 (64%) |
| Men | 23 (43%) | 13 (50%) | 10 (36%) |
| Marital Status | | | |
| Single | 22 (41%) | 11 (42%) | 11 (39%) |
| Married | 32 (59%) | 15 (58%) | 17 (61%) |
| Disease activity | | | |
| Active | 35 (65%) | 19 (73%) | 16 (57%) |
| Remission | 19 (35%) | 7 (27%) | 12 (43%) |
| Duration of IBD (year) | 4.7 ± 4.2 | 5.4 ± 4.7 | 4.1 ± 3.7 |
| BMI (kg/m2) | 23.6 ± 4.3 | 23.0 ± 5.0 | 24.1 ± 3.5 |
| Alcohol Consumption | 7 (13%) | 2 (8%) | 5 (18%) |
| Smoking | 5 (9%) | 3 (11%) | 2 (7%) |
| Educational Status | | | |
| Illiterate and high school diploma | 18 (33%) | 8 (31%) | 10 (36%) |
| Bachelor's degree | 23 (43%) | 11 (42%) | 12 (43%) |
| Master's degree | 11 (20%) | 7 (27%) | 4 (14%) |
| Doctoral degree | 2 (4%) | - | 2 (7%) |
| Past IBD-related surgery | 9 (17%) | 7 (27%) | 2 (7%) |
| Job | | | |
| Employee | 19 (36%) | 9 (35%) | 10 (36%) |
| Self-employed | 13 (24%) | 7 (27%) | 6 (21%) |
| Housekeeper | 14 (26%) | 3 (11%) | 11 (39%) |
| Unemployed | 4 (7%) | 4 (15%) | - |
| Retired with pension | 4 (7%) | 3 (11%) | 1 (4%) |

CD, crohn's disease; N, Number; UC, Ulcerative colitis; BMI, Body Mass Index.

**Table 4. Internal consistency and test-retest reliability of the Persian version of IBD-F questionnaire.**

| Questionnaire / Test | Cronbach's α coefficient (95% CI) | | ICC (95%CI) |
|---|---|---|---|
| | Test (first measurement) | Re-test (after 14 days) | Test and Retest |
| First part: q1-q5 | 0.845 (0.769–0.902) | 0.793 (0.690–0.869) | 0.904 (0.834–0.944) |
| Second part: q6-q35 | 0.963 (0.948–0.976) | 0.876 (0.823–0.919) | 0.922 (0.865–0.955) |
| Whole questionnaire * | 0.964 (0.949–0.976) | 0.888 (0.840–0.927) | 0.921 (0.865–0.954) |

ICC, intraclass correlation coefficients; CI, confidence interval.

* The whole questionnaire refers to the assembled parts I and II questions (q1-q35).

5-aminosalicylic acid, were the subjects' most prevalent medications (50% and 48%, respectively).

After 14 days, all patients were recalled to the outpatient clinic for the re-test stage. None of the participants were excluded since no changes associated with disease activity were observed within the interval between the test and re-test.

**3 Internal consistency.** Cronbach's α coefficient calculation examined internal consistency for parts I, II, and the whole instrument (assembled parts I and II questions (Q1-Q35)) (Table 4). The result demonstrated proper internal consistency, as Cronbach's α coefficient was obtained at 0.964 (0.845 first part and 0.963 second part) for the test and 0.888 (0.793 first part and 0.876 second part) for the re-test.

**4 Reproducibility.** Replies to the items were compared via test-retest reliability using the ICC. Agreement between test and re-test replies, whether for individual items or each section, was considered reproducibility Tables 4 and 5. In the first section, one item (Q3) showed moderate reliability (ICC 0.559); however, the rest owned good reliability (ICC range 0.7–0.9). In section two, one-third of all the items possess ICC between 0.5–0.75, indicating moderate reliability (Q15, Q17, Q19-21, Q23, Q24, Q26, Q28, and Q29), while others possessed satisfactory reliability (ICC range 0.7–0.9). Ultimately, ICC values between the test and re-test for the first and second part and the whole questionnaire were obtained at 0.904, 0.922, and 0.921, respectively.

## Discussion

Fatigue, the principal consequence of IBD, can negatively interfere with the patients' life, specifically in their QoL, which impresses psychological well-being and daily functioning. Fatigue assessment provides the opportunity to identify the cases vulnerable to exhaustion and those with impaired daily life; furthermore, clinicians can be informed how much their prescribed intervention to attenuate the fatigue was effective. The proper intervention for fatigue management is a determinative component for enhancing QoL, life expectancy, and treatment adherence, providing some other profits [31]. Although there are questionnaires like BFI, MFI, and FACIT-F, which generally evaluate fatigue, IBD-F is a specific-itemized questionnaire that comprehensively targets fatigue in each QoL domain. As the cross-cultural adaption of the IBD-F Persian version questionnaire, the present investigation facilitates future research on fatigue within IBD patients.

The strength of the IBD-F scale lies in the firm and rough scale development process [15]. It works in a self-assessment report manner that more accurately reflects the severity of fatigue and its influence on individuals' lives. Moreover, the scale is detailed for IBD patients, which in the first part measures the amount and intensity of fatigue, and in the second part, it can show the fatigue effects on each sector of QoL. This feature provides the opportunity to design the intervention precisely for the activity influenced mainly by fatigue. Likewise, the third part,

**Table 5. Responders agree about test and retest questions.**

| Question number | Question phrase | ICC (95% CI) |
|---|---|---|
| Section one | | |
| 1 | What is your fatigue level right now | 0.854 (0.748–0.915) |
| 2 | What was your highest fatigue level in the past two weeks | 0.888 (0.807–0.935) |
| 3 | What was lowest fatigue level in the past two weeks | 0.599 (0.309–0.767) |
| 4 | What was your average fatigue level in the past two weeks | 0.782 (0.624–0.874) |
| 5 | How much of your waking time have you felt fatigued in the past two weeks | 0.861 (0.760–0.919) |
| Section two | | |
| 1 | I had to nap during the day because of fatigue | 0.750 (0.569–0.855) |
| 2 | Fatigue stopped me from going out to social events | 0.875 (0.785–0.928) |
| 3 | I was not able to go to work or college because of fatigue | 0.866 (0.769–0.922) |
| 4 | My performance at work or education was affected by fatigue | 0.839 (0.723–0.907) |
| 5 | I had problems concentrating because of fatigue | 0.884 (0.800–0.933) |
| 6 | I had difficulty motivating myself because of fatigue | 0.855 (0.802–0.933) |
| 7 | I could not wash or dress myself because of fatigue | 0.780 (0.621–0.872) |
| 8 | I had difficulty with walking because of fatigue | 0.860 (0.759–0.919) |
| 9 | I was unable to drive as much as I need to because of fatigue | 0.928 (0.876–0.958) |
| 10 | I was not able to do as much physical exercise as I wanted to because of fatigue | 0.842 (0.728–0.908) |
| 11 | I had difficulty continuing with my hobbies or interests because of fatigue | 0.881 (0.796–0.931) |
| 12 | My emotional relationship with my partner was affected by fatigue | 0.882 (0.796–0.931) |
| 13 | My sexual relationship with my partner was affected by fatigue | 0.874 (0.783–0.927) |
| 14 | My relationship with my children was affected by fatigue | 0.838 (0.720–0.906) |
| 15 | I was low in mood because of fatigue | 0.741 (0.553–0.850) |
| 16 | I felt isolated because of fatigue | 0.840 (0.725–0.907) |
| 17 | My memory was affected because of fatigue | 0.735 (0.544–0.846) |
| 18 | I made mistakes because of fatigue | 0.779 (0.619–0.872) |
| 19 | Fatigue made me irritable | 0.602 (0.314–0.769) |
| 20 | Fatigue made me frustrated | 0.668 (0.428–0.807) |
| 21 | I got words mixed up because of fatigue | 0.691 (0.468–0.821) |
| 22 | Fatigue stopped me from enjoying life | 0.857 (0.754–0.917) |
| 23 | Fatigue stopped me from having a fulfilling life | 0.717 (0.513–0.836) |
| 24 | My self-esteem was affected by fatigue | 0.634 (0.369–0.788) |
| 25 | Fatigue affected my confidence | 0.848 (0.737–0.912) |
| 26 | Fatigue made me feel unhappy | 0.653 (0.402–0.799) |
| 27 | I had difficulties sleeping at night because of fatigue | 0.813 (0.678–0.892) |
| 28 | Fatigue affected my ability to do normal household activities | 0.637 (0.374–0.789) |
| 29 | I had to ask others for help because of fatigue | 0.716 (0.511–0.835) |
| 30 | Quality of my life was affected by fatigue | 0.758 (0.583–0.860) |

ICC, Intraclass correlation coefficient; CI, Confidence interval.

an open-ended section, identifies other confounding factors and the most fruitful actions from the participant's point of view, prioritizing solutions for analysts. Although the questionnaire seems lengthy (3 pages and forty items), none of the candidates didn't decline the invitation, and the mean time of fulfillment has been estimated at ten minutes. However, questionnaires possess some limitations; for instance, no boundary was defined as a threshold for measuring fatigue. This needs to be clarified how much fatigue level is acceptable and who potentially needs medical consideration. Factors related to fatigue, including sleep disorders, medications, psychological disorders, and dietary habits [32], effects the prevalence and severity of fatigue and have not been considered in this questionnaire [18].

Followed by the first IBD-F questionnaire development [15], cross-cultural investigations in the diverse setting have been accomplished [33–36], which all have differences from the current Persian version. The mean age of current participants (36.5 years) is more realistic and closer to IBD-patients' society [37] than other versions. Like Polish and Mexican versions of IBD-F, men and women had nearly equal participation; however, the study population in other (Brazilian, Danish, and English) versions mainly encompassed women. Patients with UC were the dominant group in the Mexican design of IBD-F; meanwhile, CD patients prevailed in Brazilian, polish, and English ones. Current investigation plus Danish design demonstrated the somehow equal contribution of IBD-sub types. In Danish, Brazilian, and Mexican IBD-F types, patients in the remission phase mostly took part; nevertheless, the current project, resembling the original English version, was carried out primarily in patients with active disease. These statistics indicate that the study population in the current version is more representative of the IBD community. It should be noted that opposed to other versions, all participants in the test phase attended the re-test phase, indicating the unity of the study population. Furthermore, participants were clients of the Gastroenterology clinic and were enrolled based on a definite diagnosis of IBD. However, in the original English version, participants for phases 1–4 volunteered in response to an advertisement newsletter, and for phase 5, participants were randomly selected from Charity's membership database [15]. Besides, this project has examined the most comorbidities and sociodemographic variables, like COVID-19, iron deficiency anemia, or educational status, which all can impress the fatigue level of IBD patients. All of these facts can make the current project distinguishable and more reliable.

Internal consistency was achieved excellent for the test (first section = 0.845, second section = 0.963, and whole questionnaire = 0.964) and good for re-test phases (first section = 0.793, second section = 0.876, and whole questionnaire = 0.888). The result so resembles the polish (first section = 0.883, second section = 0.966, and entire questionnaire = 0.968) and Mexican versions (first section = 0.87 and second section = 0.94); however, the best outcome is dedicated to the English version (first section = 0.91 and second section = 0.98). ICC values illustrate excellent reliability (first section = 0.904, second section = 0.922, and entire questionnaire = 0.921). In comparison, the result was more satisfactory than English (first section = 0.74 and second section = 0.83) and Polish versions (first section = 0.77 and second section = 0.84); however, the best result is devoted to the Brazilian version (first section = 0.92, second section = 0.97, and entire questionnaire = 0.97).

This questionnaire can be used by native Persian researchers and physicians dealing with IBD patients or specifically conduct research on fatigue in these patients. More than 2000 gastroenterologists are members of the Iranian Association of Gastroenterology and Hepatology (IAGH); IAGH reports that one-third of these specialists perform primary care and research in the field of IBD [38]. Furthermore, a domestic search engine, available at the address https://esid.research.ac.ir, searches among the faculty member who used precise MeSH terms/keywords in their research. It reports that 1107 and 3050 Iranian faculty members have

investigated about IBD and fatigue, respectively; Meanwhile, only 27 of them have investigated fatigue among IBD patients (using both keywords of fatigue and IBD in their published article). These status illustrates how much fatigue is essential and potent for investigation; the low quantity of domestic research projects in the field of fatigue in IBD patients can be attributed to the lack of Persian transcultural fatigue-scaling questionnaires for IBD patients.

The Persian version of the IBD-F questionnaire, the first valid and reliable instrument for measuring fatigue in IBD patients in Iranian society, can facilitate recognizing the patients with the most impaired activity and quality of life (those who gain higher scores in the IBD-F questionnaire) for prioritizing purposes. Furthermore, IBD-F inquires wide range of possible actions (30 questions like educational, sexual, familial, social, and habitual activities) that may be impressed by fatigue; thus, an opportunity is provided for interventionists to figure out which activity mostly impressed for designing an intervention (pharmaceutical, psychosocial, and dietary). Besides, measuring the efficacy of an intervention for attenuating fatigue (decreasing the IBD-F questionnaire score) is possible. Further controlled trials are recommended to figure out how much the IBD-F questionnaire is practical to demonstrate changes with the intervention. Besides, a need for an updated version of IBD-F in a more concise format with determined thresholds is felt.

This study reached its goal of evaluating the cross-cultural adaptation of the Persian version of the IBD-F questionnaire; however, there were also a few limitations. According to the research question and the available data, the classical psychometrics approach was applied for transcultural adaptation of the questionnaire to explore the underlying factor of scale structure. In the subsequent investigations, a method like Rasch analysis, which considers respondents' different personal traits, recommended be used to exhibit the relationship between each item's responses. It should be mentioned that considering the ethnic geography in Iran, this questionnaire probably does not provide the expected efficiency in referring patients with non-common ethnicity, culture, and language. Furthermore, the low ratio of the sample size (54 patients) to the number of questions (35 questions) may have affected the construct validity analysis.

## Conclusions

As no cross-cultural adaptation investigation of the IBD-F questionnaire in Iranian society was available, current perusal aimed to appraise the psychometrics properties of the Persian version of the IBD-F questionnaire. Through the validation process, all items were retained in the final version of the questionnaire. Test-retest reliability showed satisfactory reproducibility of the result. Persian version of the IBD-F questionnaire was valid and reliable; thus, it was deemed an appropriate scale to measure fatigue (severity, frequency, and effect on daily activities); however, it faces some limitations.

## Supporting information

**S1 Table. Past medical and drug history of the participants in test-retest reliability.**
(DOCX)

**S1 File. Persian version of inflammatory bowel disease-fatigue (IBD-F) self-assessment questionnaire.** This file contains the Persian version of the IBD-F questionnaire and can be used by native Persian researchers and physicians for the measurement of fatigue in IBD patients.
(DOCX)

## Acknowledgments

The authors would like to thank all the participants in this study and Czuber-Dochan et al. (authors of the original IBD-F questionnaire) for their kind cooperation.

## Author Contributions

**Conceptualization:** Mohamad Amin Pourhoseingholi, Makan Cheraghpour.

**Data curation:** Mohammad Reza Hatamnejad.

**Formal analysis:** Mohammad Reza Hatamnejad, Hamid Asadzadeh Aghdaei.

**Investigation:** Marzieh Shirvani, Hedieh Balaii, Amirhosein Koolaeian.

**Methodology:** Mohammad Reza Hatamnejad.

**Software:** Mohammad Reza Hatamnejad.

**Validation:** Mohammad Reza Hatamnejad, Makan Cheraghpour.

**Writing – original draft:** Mohammad Reza Hatamnejad, Shabnam Shahrokh, Makan Cheraghpour.

**Writing – review & editing:** Mohammad Reza Hatamnejad, Shabnam Shahrokh, Makan Cheraghpour.

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
