## [Decision Letter · Decision Letter 0]

1 Jun 2023

PONE-D-23-09001Translation and cross-cultural adaptation of the Persian version of inflammatory bowel disease-fatigue (IBD-F) self-assessment questionnairePLOS ONE

Dear Dr. Makan Cheraghpour,

Thank you for submitting your manuscript to PLOS ONE. After careful consideration, we feel that it has merit but does not fully meet PLOS ONE’s publication criteria as it currently stands. Therefore, we invite you to submit a revised version of the manuscript that addresses the points raised during the review process.

We look forward to receiving your revised manuscript.

Kind regards,

Shintaro Sagami

Academic Editor

PLOS ONE

Journal Requirements:

Additional Editor Comments:

Recently, there has been increasing attention on fatigue in IBD, and it is necessary to examine the validity of scoring when translated into various languages. It is foreseeable that an examination in Persian will also be necessary. However, I would like you to indicate how many doctors and researchers would utilize it when translated into Persian, and how much it would contribute to the evaluation of patients.

Furthermore, many fatigue-related scores have emerged, and in many clinical studies, FACIT-F has started to be used for patients with Inflammatory Bowel Disease (IBD). I would like you to further clarify why you chose FACIT-F for IBD.

When quoting from a review, please carefully confirm the details of the original literature (primary study) to ensure that there are no mistakes.

(Line 107) However, none of them have been validated in the IBD population and are not specific to the problems of patients with IBD

→This comment is not correct. Some fo them had validated. (Aliment Pharmacol Ther 2011; 34: 1328–1336) Please revise.

(Line 181) Following the Bartlett recommendation (23), which declared that the minimum sample size for method comparison investigation is 50 subjects,

→Is this true? I cannot find the these description in the cited article. Most of sample size will be affected by standard deviation.

Careless mistakes are noticeable, such as forgetting to put spaces. Please correct them.

Reviewers' comments:

Reviewer's Responses to Questions

**Comments to the Author**

1. Is the manuscript technically sound, and do the data support the conclusions?

Reviewer #1: Yes

Reviewer #2: Yes

2. Has the statistical analysis been performed appropriately and rigorously? 

Reviewer #1: Yes

Reviewer #2: Yes

3. Have the authors made all data underlying the findings in their manuscript fully available?

Reviewer #1: Yes

Reviewer #2: Yes

4. Is the manuscript presented in an intelligible fashion and written in standard English?

Reviewer #1: No

Reviewer #2: Yes

5. Review Comments to the Author

Reviewer #1: Please Please edit the folowing items:

change this sentence in Abstract:

Background: This is the first study to appraise the validity and reliability of the Persian version

of the Inflammatory bowel disease-fatigue (IBD-F) questionnaire.

Add New References.

Mention the innovation and importance of this research.

Add related references for lines 122-123, 131-132 .

In the method section add the Population and statistical sample.

Report the results of confirmatory factor analysis.

The limitations and applications of the study should be mentioned.

Reviewer #2: The manuscript is ok in the current form for publication. Although, the authors were used the classical methods of psychometric but they could use Rasch analysis for approving psychometric properties and it would be more robust than the previous one.

6. PLOS authors have the option to publish the peer review history of their article (what does this mean?). If published, this will include your full peer review and any attached files.

Reviewer #1: **Yes: **Amir Shams, Sport Science Research Institute (SSRI), Tehran, Iran.

Reviewer #2: **Yes: **Dr.Ali Dehqan

---

## [Author Response · Author response to Decision Letter 0]

24 Jun 2023

Dear Editor and reviewers,

Hope everything goes well.

Great appreciation for your time and consideration. We have applied all your recommendations to make the manuscript better and more precise. Changes have been specified with highlights. I hope the responses to the obscure parts be helpful.

Response to Journal Requirements:

1. The manuscript style was changed based on PLOS ONE style requirements.

2. The manuscript was edited for language usage, spelling, and grammar by the institutional language editing service; A copy of the manuscript showing changes (with track changes and highlights) was uploaded as the supporting information file.

3. There were no legal or ethical restrictions on sharing data publicly; therefore, we uploaded our data set into the Figshare repository, and they are available online via DOI: 10.6084/m9.figshare.23292446. Furthermore, we revised our data availability statement at the end of the manuscript (line: 333-334).

4. The ethics statement was transferred from the declaration section into methods (lines 116-118).

5. Caption for supplementary files was added at the end of manuscript (lines 497-501).

Response to Editor:

1. Comment: I would like you to indicate how many doctors and researchers would utilize it when translated into Persian, and how much it would contribute to the evaluation of patients. Answer: The number of clinicians and researchers dealing with IBD and Fatigue who benefit from this cross-cultural adaptation is explained through lines 328-339. Besides, the innovation and importance of this project get more prominent when there used to be no valid and reliable questionnaire to measure fatigue in IBD patients in Iranian society. Also, the application of this questionnaire for screening, prioritizing, designing an intervention, and measuring the efficacy of the designed intervention is explained within lines 340-351 

2. Comment: Furthermore, many fatigue-related scores have emerged, and in many clinical studies, FACIT-F has started to be used for patients with Inflammatory Bowel Disease (IBD). I would like you to further clarify why you chose FACIT-F for IBD. Answer: As you said, FACIT-F has been used in many clinical studies for patients with Inflammatory Bowel Disease (IBD); however, the reasons that IBD-F may seem to be more fruitful rather than FACIT-F for measuring fatigue are explained through lines 103-110 (designed explicitly for IBD patients, being more comprehensive, having more extended recall period, and owning a counseling section). 

3. Comment: Please revise this sentence: However, none of them have been validated in the IBD population and are not specific to the problems of patients with IBD. Answer: Thanks for your valuable comment. FACIT and Fatigue questionnaires are validated for IBD patients; however, they comprise 13 and 20 items, respectively, to measure the severity and impact of fatigue on daily activities. IBD-F is a 35-item scale for measuring fatigue, so it can more comprehensively evaluate the severity and effects of fatigue. Thus, the previous sentence (lines 105 - 107) was revised to " Some of them have been validated in the IBD population; however, there was a need for a more comprehensive questionnaire that examined more precisely the effect of fatigue on the daily activity of IBD patients. "

4. Comment: Is this sentence true (Following the Bartlett recommendation (23), which declared that the minimum sample size for method comparison investigation is 50 subjects)? I cannot find this description in the cited article. Answer: As we figured out that mentioned sentence caused the misunderstanding, it was omitted, and the calculation of the sample size via G power version 3.1.9 was replaced (alpha-error: 0.05, beta-error: 0.05, power: 0.95, and effect size: 0.83 (based on correlation of the original IBD-F questionnaire)) (lines 184-187).

5. Comment: Careless mistakes are noticeable, such as forgetting to put spaces. Please correct them. Answer: The manuscript underwent the language editing services by institutional language editing service and all the efforts were made to correct the writing errors.

Response to reviewer 1: 

1. Comment: Change this sentence in Abstract: Background: This is the first study to appraise the validity and reliability of the Persian version of the Inflammatory bowel disease-fatigue (IBD-F) questionnaire. Answer: Your mentioned sentence in the abstract (lines 30-31) was changed to " This study appraises the psychometric properties of the Persian version of the Inflammatory bowel disease-fatigue (IBD-F) questionnaire."

2. Comment: Mention the innovation and importance of this research. Answer: The number of clinicians and researchers dealing with IBD and Fatigue who benefit from this cross-cultural adaptation is explained through lines 328-339. Besides, the innovation and importance of this project get more prominent when there used to be no valid and reliable questionnaire to measure fatigue in IBD patients in Iranian society.

3. Comment: Add related references for lines 122-123, 131-132. Answer: A reference for lines (120-121, previous line number 122-123) was added that illustrates how the standard forward and backward was carried out (doi: 10.1111/j.1524-4733.2005.04054.x. PubMed PMID: 15804318); furthermore, for lines (128-129, previous line numbers 131-132) we demonstrated that mean of " compared with the original issue " was the first version of IBD-F questionnaire (doi: 10.1016/j.crohns.2014.04.013. PubMed PMID: 24856864).

4. Comment: In the method section add the Population and statistical sample. Answer: Calculation of sample size via G* Power software from the IBD patients’ population was explained within lines 184-185.

5. Comment: Report the results of confirmatory factor analysis. Answer: Other trans-cultural versions of the IBD-F questionnaire (like Polish doi: 10.5114/pg.2021.106665, and Danish PubMed PMID: 28869029) did not undergo the CFA because they evaluated the validity and reliability without any dimension reduction (omitting some items). We acted exactly like them; however, based on your recommendation, CFA was done, via LISREL version 8, to evaluate the construct validity within lines 177-181 and 236-238. The results show the relative property (borderline results) of observed data to the model fitness; however, the low ratio of sample size (54 patients) to the number of items (35 questions) may affect the construct validity analysis, and with a more comprehensive sample size a finer result of fitness will be achieved; thus, this issue cannot be directly related to the questionnaire, and it was because of limitation in methodology (sample size).

6. Comment: The limitations and applications of the study should be mentioned. Answer: The application of this questionnaire for screening, prioritizing, designing an intervention, and measuring the efficacy of the designed intervention is explained within lines 340-351. Also, the current study's limitations were mentioned in lines 352-361.

Response to reviewer 2: 

Comment: The manuscript is ok in the current form for publication. Although, the authors were used the classical methods of psychometric but they could use Rasch analysis for approving psychometric properties and it would be more robust than the previous one. Answer: Thanks for your precious comment. Classical psychometric and Rasch analysis are two different approaches used in the trans-cultural adaptation of a questionnaire. Classical methods are used to explore the underlying factor structure of a questionnaire. In contrast, Rasch analysis is used to model the relationship between a person's responses to the questionnaire items and the latent trait being measured. Rasch analysis is a newer method for evaluating transcultural adaption; however, both approaches have their strengths and weaknesses, and the choice of which method to use depends on the research question and the available data. We recommended that in a subsequent analysis article, the Rasch analysis can be performed to evaluate the transcultural adaption of the questionnaire (lines 355-357).

Makan Cheraghpour, Ph.D.

Shabnam Shahrokh MD.

---

## [Decision Letter · Decision Letter 1]

29 Jun 2023

Translation and cross-cultural adaptation of the Persian version of inflammatory bowel disease-fatigue (IBD-F) self-assessment questionnaire

PONE-D-23-09001R1

Dear Dr. Cheraghpour,

We’re pleased to inform you that your manuscript has been judged scientifically suitable for publication and will be formally accepted for publication once it meets all outstanding technical requirements.

Kind regards,

Shintaro Sagami

Academic Editor

PLOS ONE

Additional Editor Comments (optional):

Reviewers' comments:

Reviewer's Responses to Questions

**Comments to the Author**

1. If the authors have adequately addressed your comments raised in a previous round of review and you feel that this manuscript is now acceptable for publication, you may indicate that here to bypass the “Comments to the Author” section, enter your conflict of interest statement in the “Confidential to Editor” section, and submit your "Accept" recommendation.

Reviewer #1: All comments have been addressed

2. Is the manuscript technically sound, and do the data support the conclusions?

Reviewer #1: Yes

3. Has the statistical analysis been performed appropriately and rigorously? 

Reviewer #1: Yes

4. Have the authors made all data underlying the findings in their manuscript fully available?

Reviewer #1: No

5. Is the manuscript presented in an intelligible fashion and written in standard English?

Reviewer #1: Yes

6. Review Comments to the Author

Reviewer #1: (No Response)

7. PLOS authors have the option to publish the peer review history of their article (what does this mean?). If published, this will include your full peer review and any attached files.

Reviewer #1: **Yes: **Amir Shams, Sport Sciences Research Institute of Iran

---

## [Editor Report · Acceptance letter]

14 Jul 2023

PONE-D-23-09001R1 

Translation and cross-cultural adaptation of the Persian version of inflammatory bowel disease-fatigue (IBD-F) self-assessment questionnaire 

Dear Dr. Cheraghpour:

I'm pleased to inform you that your manuscript has been deemed suitable for publication in PLOS ONE. Congratulations! Your manuscript is now with our production department. 

Kind regards, 

on behalf of

Dr. Shintaro Sagami 

Academic Editor

PLOS ONE